# Impact of diabetes mellitus on postoperative outcomes in individuals with non-small-cell lung cancer: A retrospective cohort study

**Teruya Komatsu**[1,2]**, Toyofumi F. Chen-Yoshikawa**[2]**, Masaki Ikeda**[2]**, Koji Takahashi**[2]**, Akiko Nishimura**[3,4]**, Shin-ichi Harashima**[3,5]**, Hiroshi Date**[2]*

1 Division of Thoracic Surgery, National Hospital Organization Nagara Medical Center, Gifu, Japan,
2 Department of Thoracic Surgery, Graduate School of Medicine, Kyoto University, Kyoto, Japan,
3 Department of Human Health Sciences, Graduate School of Medicine, Kyoto University, Kyoto, Japan,
4 Faculty of Nursing, School of Medicine, Nara Medical University, Nara, Japan, 5 Goshominami Harashima Clinic, Kyoto, Japan

* hdate@kuhp.kyoto-u.ac.jp

**Data Availability Statement:** The dataset for this study has been submitted as a supplementary file.

## Abstract

### Objectives

Studies showing that individuals with non-small cell lung cancer (NSCLC) and diabetes mellitus (DM) have reported poor outcomes after pulmonary resection with varying results. Therefore, we investigated the clinical impact of preoperative DM on postoperative morbidity and survival in individuals with resectable NSCLC.

### Patients and methods

Data of individuals who underwent pulmonary resection for NSCLC from 2000 to 2015 were extracted from the database of Kyoto University Hospital. The primary endpoint was the incidence of postoperative complications, and secondary endpoints were postoperative length of hospital stay and overall survival. The survival rate was analyzed using the Kaplan–Meier method.

### Results

A total of 2,219 patients were eligible for the study. The median age of participants was 67 years. Among them, 39.5% were women, and 259 (11.7%) presented with DM. The effect of DM on the incidence of postoperative complications and postoperative length of hospital stay was not significant. Although the 5-year survival rates were similar in both patients with and without DM (80.2% versus 79.4%; p = 0.158), those with DM who had a hemoglobin A1c level $\geq$ 8.0% had the worst survival.

### Conclusions

In individuals with resectable NSCLC, preoperative DM does not influence the acute phase postoperative recovery. However, poorly controlled preoperative DM could lead to low postoperative survival rates.

**Funding:** The authors received no specific funding for this work.

**Competing interests:** The authors have declared that no competing interests exist.

## Introduction

Lung cancer is the leading cause of cancer-related deaths worldwide [1]. Most individuals with lung cancer present with advanced disease, and the 5-year survival rate of individuals with lung cancer ranges from 11% to 46% [1–3]. Concurrently, the prevalence of diabetes mellitus (DM) is continually increasing, that is, approximately 451 million adults worldwide have DM in 2017, and this number will increase to 693 million by 2045 [4].

Considering the high prevalence of both lung cancer and DM, clinicians will soon be required to treat and manage individuals with both DM and lung cancer in clinical settings [5].

In Japan, 15.5% of men and 9.8% of women are suspected to have type 2 DM [6]. Epidemiologic evidence has indicated that individuals with DM are at a significantly high risk of cancer, such as pancreatic, hepatic, colorectal, breast, urinary tract, and endometrial cancers [7]. In contrast, decreased incidence of prostate cancer is observed in individuals with DM [7,8]. A recent meta-analysis has shown that DM might increase the risk of lung cancer, particularly among women [9]. Evidence on the correlation between DM and outcome of lung cancer is inconclusive [10–12]. The most recent large-scale cohort study of an Asian population has shown that DM was not statistically significantly associated with the risk of death from non-small cell lung cancer (NSCLC) [13]. In Japan, lung cancer is the leading cause of death in individuals with DM complicated by malignant neoplasia [14].

Surgical resection is the treatment of choice for individuals with resectable NSCLC. Although pathological tumor staging is the most powerful predictor of prognosis and survival, significant variations exist in prognosis and survival between individuals with resectable NSCLC of similar stage [15].

Studies regarding the association between DM and postoperative complications have been inconclusive [16–20]. In addition, studies about the relationship between DM and survival in individuals who undergo resection for NSCLC have reported contrasting results [15,16,21–24] (S1 Table).

Irrespective of the weak evidence on the association between DM and postoperative morbidity and survival, preoperative DM should be considered a risk factor as operations are sometimes postponed for better preoperative glycemic control. Therefore, studies about the short- and long-term effects of DM in individuals with resected NSCLC are urgently desirable. Therefore, we conducted a retrospective study to investigate the effect of DM on the postoperative morbidity and survival of individuals who underwent resection for NSCLC.

## Patients and methods

### Ethical statement

The institutional review board (IRB) of Kyoto University Hospital, Japan, approved this study (R0710). Informed consent was obtained in the form of opt-out on a dedicated website.

### Data source

This retrospective cohort study was conducted using data obtained from the electronic database of Kyoto University Hospital, which contains comprehensive health care information, including detailed medical records of ambulatory and inpatient care and diabetes conditions.

### Patient identification

We retrospectively examined all individuals diagnosed with NSCLC who underwent curative surgery at Kyoto University Hospital between May 2000 and July 2015. We excluded individuals with incomplete data.

## Exposed variables

Data on clinical characteristics, including age, sex, smoking habits, preoperative comorbidities, incision, surgery type, intraoperative bleeding, procedure time, histological type, pathological stage (P-stage), postoperative length of hospital stay, and postoperative complications were collected. Current smoking status was defined as active smoking before undergoing resection; smoking status of none was defined as having quit smoking before undergoing resection or having no smoking history. Histological classification was performed according to the fourth edition of the World Health Organization Classification of Tumors. P-stage was determined according to the tumor, node, metastases (TNM) criteria in the seventh edition of the Union for International Cancer Control Classification. In the present study, the diabetes are individuals with known diabetes or those diagnosed preoperatively as DM by board-certified DM specialists. For diabetes-related subgroup analyses, participants were categorized according to their preoperative HbA1c levels: HbA1c $< 7.0\%$, $8.0\% > $ HbA1c $\geq 7.0\%$, and HbA1c $\geq 8.0\%$.

## Outcome measures

The primary endpoint was the incidence of postoperative complications. The secondary endpoints were postoperative length of hospital stay and overall survival (OS). Postoperative complications as a primary endpoint was assessed from the end of surgery to discharge from the thoracic surgery unit. OS as a secondary endpoint was calculated from the date of surgery to the final event (death or loss to follow-up).

Details of postoperative complications were recorded (cerebrovascular, cardiovascular, pulmonary, and pleural complications, bacterial infections, and others). Moreover, postoperative complications were graded according to the extended Clavien–Dindo classification of surgical complications, which was established by the Japan Clinical Oncology Group [25]: grade I, conditions requiring clinical observation only, which includes the use of medications, such as antiemetics, antipyretics, analgesics, and diuretics; grade II, conditions requiring medical management (e.g., antibiotics or antiarrhythmic drugs); grade IIIa, conditions requiring medical intervention under local anesthesia (e.g., bronchoscopic aspiration or pleurodesis); grade IIIb, conditions requiring surgical intervention under general anesthesia (e.g., re-operation); grade IVa, life-threatening complications requiring intensive care unit management (e.g., mechanical ventilation); grade IVb, life-threatening complications involving multiple organ failure; and grade V, death.

## Statistical analysis

Continuous variables were presented as medians and interquartile ranges (IQRs) and categorical variables as number (%). Univariate comparisons were performed using the Pearson chi-square test for categorical variables and Wilcoxon rank-sum test for continuous variables. The Kaplan–Meier method was used to estimate OS. Survival times were calculated from the initial event (date of surgery) to the final event (death or loss to follow-up). A two-sided p-value of $< 0.05$ was considered statistically significant. Cox proportional hazards regression was performed to estimate hazard ratios and 95% confidence intervals (CI) for factors associated with survivals. Statistical analyses were performed using the R software/environment. R is an open source project distributed under the GNU General Public License (Copyright 2007, Free Software Foundation, Inc., http://www.gnu.org/licenses/gpl.html). At the time of writing this manuscript, R-2.13.1 was available.

This study conforms to the Strengthening the Reporting of Observational Studies in Epidemiology guidelines.

## Results

### Baseline characteristics of participants

Between 2000 and 2015, 2242 lung resections for NSCLC had been performed. A total of 23 patients had incomplete data, and data of 2219 (99.0%) patients were reviewed. Baseline demographic characteristics of participants are shown in Table 1.

The median age of participants was 69 (IQR; 61–74) years, and among the participants, 60.7% were men. Current smokers accounted for 28.8% of the sample population. The median follow-up period was 39.0 months, and 97% of participants were followed up to death or at least 5 years. Among the individuals with NSCLC, 259 (11.7%) presented with DM, whereas

**Table 1. Characteristics of the 2219 patients (Non-diabetes vs diabetes) including Post-operative stay and complications.**

| | Non-diabetes (n = 1960) | Diabetes (n = 259) | P value |
|---|---|---|---|
| Age (years) | 68 (61–74) | 70 (65–75) | <0.001 |
| Gender | | | <0.001 |
| Male | 1161 (59.3%) | 187 (72.2%) | |
| Female | 798 (40.7%) | 72 (27.8%) | |
| Smoking status | | | 0.429 |
| Current | 559 (28.5%) | 80 (30.8%) | |
| None | 1401 (71.5%) | 179 (69.2%) | |
| HbA1c level (%) | 5.6 (5.4–6.4) [a] | 6.4 (6.2–7.6) | 0.003 |
| Preoperative morbidities | | | |
| CAD | 100 (5.1%) | 30 (11.6%) | <0.001 |
| Arrhythmia | 47 (2.4%) | 14 (5.4%) | 0.005 |
| CVD | 101 (5.2%) | 21 (8.1%) | 0.049 |
| COPD | 109 (5.6%) | 21 (8.1%) | 0.100 |
| Incision | | | 0.354 |
| Thoracotomy | 662 (33.8%) | 80 (30.9%) | |
| VATS | 1298 (66.2%) | 179 (69.1%) | |
| Surgery type | | | 0.052 |
| Less than lobectomy | 572 (29.2%) | 91 (35.1%) | |
| Lobectomy or more | 1388 (70.8%) | 168 (64.9%) | |
| Intraoperative bleeding (ml) | 60 (5–160) | 50 (5–132) | 0.127 |
| Operative time (min.) | 210 (167–263) | 216 (171–261) | 0.907 |
| Pathology | | | <0.001 |
| Adenocarcinoma | 1403 (71.6%) | 161 (62.2%) | |
| Squamous cell | 427 (21.8%) | 80 (30.9%) | |
| Others | 130 (6.6%) | 18 (6.9%) | |
| Pathological stage (I/II/III/IV) | 1452/223/259/26 (74.0%/11.8%/13.2%/1.0%) | 191/36/28/4 (73.7%/13.9%/10.8%/1.6%) | 0.502 |
| Post-operative complications [b] | | | 0.106 |
| None or I | 1440 (73.5%) | 178 (68.7%) | |
| II ≤ | 520 (26.5%) | 81 (31.3%) | |
| Post-operative stay (days) | 13 (9–19) | 12 (9–17) | 0.058 |

[a] The number of patients without DM whose HbA1c levels were measured was small (n = 84).

[b] The extended Clavien-Dindo classification of surgical complications established by the Japan Clinical Oncology Group was used for grading post-operative complications.

CAD: Coronary artery disease CVD: cerebrovascular disease.

COPD: Chronic obstructive pulmonary disease DM: diabetes mellitus.

VATS: Video-assisted thoracoscopic surgery.

1960 (88.3%) did not. Smoking status, incision, type of surgery, intraoperative bleeding, operative time, chronic obstructive pulmonary disease as a preoperative comorbidity, and pathological stage did not differ among the diabetes and non-diabetes groups (Table 1). The occurrence of comorbidities, such as coronary artery disease, arrhythmia, and cerebrovascular disease (CVD), was significantly higher in individuals with DM than in those without DM (Table 1).

## Primary endpoint

The incidence of postoperative complications was comparable between individuals with DM and those without DM (Table 1). In the subgroup analysis of HbA1c levels (HbA1c levels < 7.0%, ≥ 7.0%, and ≥ 8.0%), the incidence of postoperative complications did not significantly differ among the three subgroups (Table 2). These findings were also verified by adjusted estimates (Table 3).

## Secondary endpoints

The postoperative length of hospital stay was comparable between individuals with DM and those without DM (Table 1). In the subgroup analysis of HbA1c levels (HbA1c levels < 7.0%, ≥ 7.0%, and ≥ 8.0%), the postoperative length of hospital stay did not significantly differ among the three subgroups (Table 2).

Post-surgical survival did not differ significantly between patients with and without DM (Fig 1, 5-year survival rate: 80.2% vs 79.4%, p = 0.158).

Cox regression analysis showed the similar result that DM were not an independent predictor of poorer survival (hazard ratio 1.16 [95% CI 0.85–1.57], p = 0.34; Table 4).

A significant difference was observed in postsurgical survival between patients with DM who had HbA1c levels < 7.0% and ≥ 7.0% (Fig 2, 5-year survival rate: 80.5% vs 60.0%, p = 0.043).

Participants with HbA1c levels of ≥8.0% had the worst survival among the three subgroups (Fig 3).

NSCLC-related death occurred in 86.2%, 80.0%, and 85.7% of participants with DM who had HbA1c levels < 7.0%, 7.0 ≤ HbA1c < 8.0, and ≥ 8.0%, respectively (p = 0.892).

## Discussion

DM has been associated with increased risks for several types of cancer, including lung cancer [22]. There have been conflicting reports regarding the association between DM and

**Table 2. Post-operative stay and complications of DM patients stratified by HbA1c levels.**

|  | HbA1c (%) < 7.0 (n = 180) | 7.0 ≤ HbA1c < 8.0 (n = 57) | 8.0 ≤ HbA1c (n = 22) | P value |
|---|---|---|---|---|
| Post-operative complications |  |  |  | 0.09 |
| None or I | 121 (67.2%) | 37 (64.9%) | 19 (86.4%) |  |
| II ≤ | 59 (32.8%) | 20 (35.1%) | 3 (13.6%) |  |
| Post-operative complications |  |  |  | 0.11 |
| Cerebrovascular | 1 (0.5%) | 0 (0%) | 0 (0%) |  |
| Cardiovascular | 10 (5.6%) | 1 (1.8%) | 1(4.5%) |  |
| Pulmonary and pleural | 42 (23.3%) | 11 (19.3%) | 4 (18.2%) |  |
| Bacterial infection | 10 (5.6%) | 5 (8.8%) | 0 (0%) |  |
| Others | 21 (11.7%) | 0 (0%) | 3 (13.6%) |  |
| Post-operative stay (days) | 12 (9–19) | 13 (9–18) | 11 (8–17) | 0.07 |

**Table 3. Adjusted estimates of the post-operative complications.**

| Predictor variables | | Odds ratio | 95% confidence interval | P value |
|---|---|---|---|---|
| DM | | 1.29 | 0.96–1.73 | 0.07 |
| Non-diabetes | | Referent group | | |
| Diabetes | HbA1c (%) < 7.0 | 0.71 | 0.29–0.74 | 0.12 |
| | 7.0 ≤ HbA1c < 8.0 | 0.89 | 0.20–0.99 | 0.06 |
| | 8.0 ≤ HbA1c | 1.73 | 0.28–1.87 | 0.51 |

prognoses with NSCLC [10–12,23]. In addition, whether DM affects the prognosis of individuals undergoing resection for NSCLC has not yet been elucidated [10].

DM is associated with microcirculation disorders, which is considered to cause postsurgical complications [17]. A meta-analysis has revealed that DM can be an independent risk factor for bronchopleural fistula after pulmonary resection in the Asian population [18]. Another study has reported that DM was associated with an increased risk of postoperative mortality [16]. In contrast, a study that analyzed the risk factors for postoperative nosocomial pneumonia in stage I–IIIa lung cancer reported that DM was not a significant risk factor for postoperative pneumonia [19,20]. Therefore, the influence of DM on postoperative course has been inconsistent. Our results showed that preoperative DM did not have a significant negative effect on the incidence of postoperative complications and postoperative length of hospital stay. However, regardless of the effect of preoperative DM status, cautious management of individuals with DM who are undergoing resection for NSCLC is still advisable.

In terms of preoperative glycemic control for postoperative complications and postoperative length of hospital stay, a preoperative HbA1c level < 7.0% is an optimum indicator that can be used in reducing postoperative infectious complications in non-cardiac surgeries [26]. A chronic hyperglycemic state leads to impaired immune system, which contributes to the increased incidence of postoperative infections [26,27]. However, the results of the present study were not consistent with those of other studies. Neither the incidence of postoperative complications nor the postoperative length of hospital stay was affected by preoperative DM. The optimal target for preoperative glycemic control in lung cancer has not always been

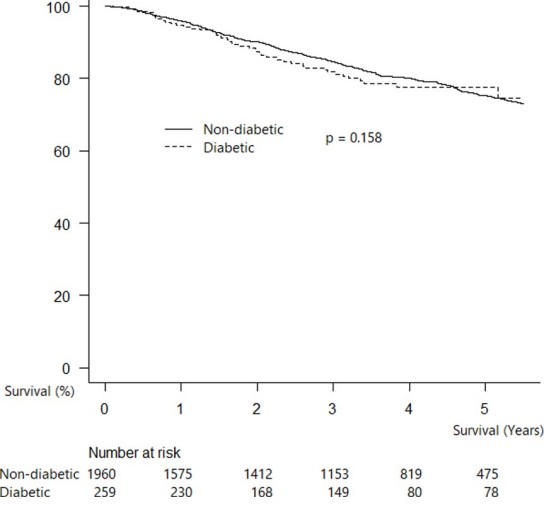

**Fig 1. Kaplan–Meier survival probability curve for patients with and without DM after NSCLC resection (adjusted for age and gender).**

**Table 4. Multivariate Cox regression of survivals in patients of resected NSCLC.**

|  | Hazard ratio | 95% confidence interval | P value |
|---|---|---|---|
| Age (years) | 1.02 | 1.01–1.03 | <0.001 |
| Gender | 2.85 | 1.28–6.34 | 0.01 |
| Preoperative DM | 1.16 | 0.85–1.57 | 0.34 |
| Preoperative morbidities |  |  |  |
| CAD | 1.02 | 0.67–1.55 | 0.30 |
| Arrhythmia | 1.04 | 0.98–1.22 | 0.72 |
| Surgery type |  |  |  |
| Less than lobectomy | 1.13 | 0.90–1.41 | 0.26 |
| Lobectomy or more | 0.99 | 0.89–1.01 | 0.88 |
| Pathology |  |  |  |
| Adenocarcinoma | 1.30 | 1.11–1.47 | 0.61 |
| Squamous cell | 1.67 | 1.44–2.02 | 0.52 |
| Others | 2.41 | 1.79–3.23 | 0.02 |

defined, and strict preoperative glycemic control might be time-consuming, leading to delayed surgical treatment for NSCLC. Considering the study that demonstrated that patients with poorly controlled DM are already known to be at increased risk for morbidity and mortality during the long-term follow-up, it is well expected that poorly controlled DM patients would fare worse in any setting when compared to patients with better glycemic control. As a result, it cannot be concluded that delaying surgery to correct HbA1c will improve surgical outcomes [28,29]. Our results also indicated that it is not necessary to postpone radical operation for NSCLC even though preoperative DM control for preventing diabetes-associated cardiovascular and microvascular complications is not achieved; therefore, proceeding with surgery for NSCLC with simultaneous DM management is advisable.

In contrast, better glycemic control might be helpful for the postsurgical survival of individuals with resected NSCLS. Although preoperative DM was not found to be associated with poor OS, our data showed that individuals with HbA1c levels ≥ 7.0% had worse postsurgical

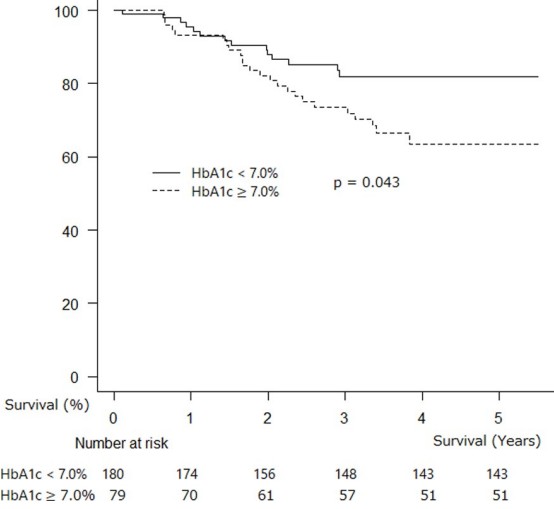

**Fig 2. Kaplan–Meier survival probability curve for postsurgical patients with HbA1c < 7.0% and HbA1c ≥ 7.0% (adjusted for age and gender).**

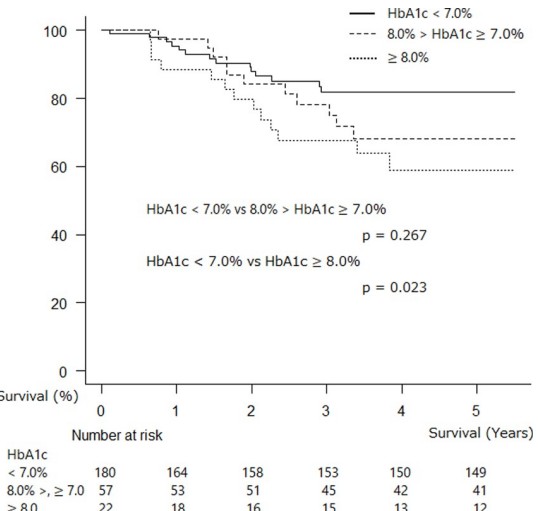

**Fig 3. Kaplan–Meier survival probability curve for postsurgical patients with HbA1c < 7.0%, 7.0 ≤ HbA1c < 8.0%, and HbA1c ≥ 8.0% (adjusted for age and gender).**

survival than those with HbA1c levels < 7.0%; meanwhile, those with HbA1c levels ≥ 8.0% had the worst survival. To achieve improved survival, proper glycemic control (HbA1c levels < 7%) is recommended for individuals with DM undergoing lung cancer operations [30]. In contrast, DM has no effect on the survival of individuals undergoing resection for stage I NSCLC [22] and long-term (60-month) survival [21]. Considering that the number of NSCLC-related deaths was similar in the three subgroups in the present study (86.2%: HbA1c < 7.0%, 80.0%: 7.0 ≤ HbA1c < 8.0, and 85.7%: HbA1c ≥ 8.0%; p = 0.892) and that high HbA1c levels were associated with poor survival, the effect of DM control status on survival in individuals with resected NSCLC is undoubtedly significant. However, optimal glycemic control of DM in individuals with operable NSCLC should be addressed in the future.

This study had several limitations. First, this was a retrospective study. Second, the period during which data were collected might have influenced the demographic characteristics of patients, operative techniques, and perioperative management. Third, the severity and duration of DM and types of diabetes therapy were not examined. In the present study, the number of individuals with DM who had HbA1c levels ≥ 8.0% was low to investigate the negative impact of preoperative poor glycemic control on the prognosis of resected NSCLC.

In conclusion, the incidence of postoperative complications and postoperative length of hospital stay were not affected by preoperative DM in individuals with NSCLC. In contrast, survival was affected by status of DM control, particularly in individuals with HbA1c levels > 8.0% who have resected NSCLC. Thus, further studies must be conducted to confirm whether optimal preoperative glycemic control is achieved.

## Supporting information

**S1 Table. The included studies regarding the effect of DM on survivals in patients managed surgically for NSCLC.**
(DOCX)

**S1 File.**
(XLS)

## Author Contributions

**Conceptualization:** Teruya Komatsu, Koji Takahashi, Akiko Nishimura.

**Data curation:** Teruya Komatsu, Masaki Ikeda, Koji Takahashi.

**Formal analysis:** Teruya Komatsu, Masaki Ikeda, Koji Takahashi, Akiko Nishimura, Shin-ichi Harashima.

**Investigation:** Teruya Komatsu, Koji Takahashi, Akiko Nishimura.

**Project administration:** Toyofumi F. Chen-Yoshikawa.

**Supervision:** Toyofumi F. Chen-Yoshikawa, Shin-ichi Harashima, Hiroshi Date.

**Writing – original draft:** Teruya Komatsu.

**Writing – review & editing:** Toyofumi F. Chen-Yoshikawa, Akiko Nishimura, Shin-ichi Harashima.

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
