## [Decision Letter · Decision Letter 0]

20 Aug 2020

PONE-D-20-19563

Impact of diabetes mellitus on postoperative outcomes in individuals with non-small-cell lung cancer: a retrospective cohort study

PLOS ONE

Dear Dr. Komatsu,

Thank you for submitting your manuscript to PLOS ONE. After careful consideration, we feel that it has merit but does not fully meet PLOS ONE’s publication criteria as it currently stands. Therefore, we invite you to submit a revised version of the manuscript that addresses the points raised during the review process.

Please response the problems pointed out by reviewers.

We look forward to receiving your revised manuscript.

Kind regards,

Yoshiaki Taniyama, MD, PhD

Academic Editor

PLOS ONE

Journal Requirements:

Additional Editor Comments (if provided):

The author investigated the clinical impact of preoperative DM on postoperative morbidity and survival in individuals with resectable NSCLC. The effect of DM on the incidence of postoperative complications and postoperative length of hospital stay was not significant. Patients with DM who had a hemoglobin A1c level ≥ 8.0% had bad survival in 5-year survival rates.

1. The author showed that 5-year survival rates is similar in patients between with or without DM. Is there difference in 5-year survival rates between with or without insulin use?

2. Is there possibility that insulin may effect on the NSCLC growth?

3. Are there any DM drugs which may improve or worse the 5-year survival rates?

Reviewers' comments:

Reviewer's Responses to Questions

**Comments to the Author**

1. Is the manuscript technically sound, and do the data support the conclusions?

Reviewer #1: No

2. Has the statistical analysis been performed appropriately and rigorously? 

Reviewer #1: No

3. Have the authors made all data underlying the findings in their manuscript fully available?

Reviewer #1: No

4. Is the manuscript presented in an intelligible fashion and written in standard English?

Reviewer #1: Yes

5. Review Comments to the Author

Reviewer #1: The study by Komatsu et al seeks to evaluate whether patients with diabetes undergoing resection for NSCLC have a higher risk of postoperative complications and poor overall survival compared to those without DM undergoing the same procedure. They conducted a retrospective cohort study leveraging information collected in a clinical database from patients with NSCLC who underwent surgery at Kyoto University Hospital between 2000 and 2015.

The research question is likely to be of interest to clinicians, surgeons, patients, and the results are potentially useful for counseling and clinical management. However, the manuscript is lacking important methodological details and the analyses will need revision in order to make valid conclusions. The manuscript would also benefit from some editing.

Described below are some methodological details that require clarification or revisions and other suggestions as well:

1. Recommend shortening the Introduction section as several details are repeated in the Discussion. A few sentences could also benefit from editing to improve clarity (e.g., lines 47-48, line 62, line 69).

Line 82: Hopefully "no effect" is not just based on statistical significance. I think it would be helpful to include a table (perhaps under Supplementary section) that summarizes the observed findings from references 15, 16, 21-24.

2. Patient Identification (Lines 104-105): “We retrospectively examined all individuals diagnosed with NSCLC who underwent curative surgery at Kyoto University Hospital between May 2000 and July 2015. We excluded individuals with incomplete data.”

How many eligible subjects from the cohort were excluded due to incomplete data? Based on the number, this poses a potential threat to selection bias. Please describe the proportion of missingness in the primary variables of interest. Were excluded individuals similar to those included?

3. Given the observational nature of this study, did the researchers consider matching exposed (DM) with unexposed (no DM) on important confounders such as age, gender, stage, histological type? Statistical adjustment is okay, but matching in the design stage would give better control of these confounders. Also, unclear why the non-diabetic group includes 17 year olds. Could they clarify?

4. Outcome measures: The authors should clarify the time-point for the primary (postoperative complications) and secondary endpoints (overall survival) [ lines 122, 123]. Also, since a database was used to ascertain information on outcome measures, what is known about the validity of the grading scale used for postoperative complications? Was “Grade” already in the database or created from information collected?

5. Statistical Analysis & Results:

(i) Line 138: while it is okay to use a median for continuous variables, to get a sense of the variability, it would be useful to report the IQR (interquartile range).

(ii) For categorical variables in all Tables, please show the frequency%, not just the frequency.

(iii) Lines 141-142: “Survival times were calculated from the date of surgery.”

The operational definition of time-to-event should include when the time starts (i.e., date of surgery) and stops (not provided).

(iv) Lines 145-147: “For cox regression, initial univariate comparisons were performed using relevant variables and those with an association yielding a p-value of < 0.1 were put into the final models.”

What is the rationale for this approach especially since prediction is not the goal of this study? The objective is to estimate the relationship between DM and overall survival and in order to get a valid estimate of this association, it should be adjusted for potential confounders. The choice of the latter should not be based on p-value <0.1.

Also, side note, the “c” in cox should be upper case (Cox)

(v) Primary endpoint: lines 178 - 182. The incorrect Table number is mentioned. Should be Table 2.

For the primary endpoint, suggest the following analysis

Analysis 1: show the risk of postoperative complications (>= Grade II vs. Grade 1/none) for non-DM and DM groups and in addition to the n and %, please show the adjusted estimates (hazard ratios, since this also likely time to event). Alternatively, if defined as a binary event, then a different statistical model could be used.

Analysis 2: Repeat Analysis 1, but show a dose-response effect by now creating 3-4 groups:

Non DM (reference group)

DM with HgA1c% <7%

DM with HgA1c% >=7% to <8%

DM with HgA1c% >=8% (worry this will not be statistically efficient due to small n, so could combine groups 3 and 4 together)

(vi) Secondary endpoints: The incorrect Table number is mentioned for length of stay (line 188). Should be Table 2.

- Since these are observational data, the Kaplan-Meier survival curves should be adjusted for important confounders (e.g., age, gender)

- Table 3: please clarify if this is multivariable. Also see comment (iv) above under Statistical analysis related to this outcome.

- The authors noted that HgA1c% was measured in a very small group of non DM subjects. So it is unclear why they would include this variable in the Cox model shown in Table 3 and will certainly be collinear with DM (yes or no variable), unless the results shown are from univariable Cox models.

6. Discussion needs editing and should be revised once the analysis have been performed correctly.

6. PLOS authors have the option to publish the peer review history of their article (what does this mean?). If published, this will include your full peer review and any attached files.

Reviewer #1: **Yes: **Rita Popat

---

## [Author Response · Author response to Decision Letter 0]

6 Oct 2020

Dear the editor and the reviewer

Thank you very much for your e-mail and review of the manuscript (PONE-D-20-19563) that we submitted on June 25, 2020. We thank the editor and the reviewer for providing constructive comments that have significantly improved the quality of the original manuscript.

Response to the editor’s comments:

1. The editor has raised important and interesting points regarding medications for DM. 

Owing to the limitation of data retrieval, we could not identify which patients with diabetes were treated with insulin. Therefore, we were unable to evaluate the difference in 5-year survival rates between patients who were treated with or without insulin.

2. The editor has pointed out that insulin may be associated with the progression of non-small cell lung cancer (NSCLC). This possibility has recently been discussed in the literature. As mentioned above, we are unable to evaluate whether insulin might have influenced NSCLC growth owing to the limitation of data retrieval. We think that analyses regarding the association between insulin and NSCLC growth is really interesting and needed. 

3. In the present study, a variety of antihyperglycemic medications were prescribed for diabetes, such as metformin, sulfonylureas, DPP-4 inhibitors, thiazolidinediones, and glucagon-like peptide agonists. These were prescribed as monotherapy or combination therapy. However, we could not collect detailed information regarding antihyperglycemics. Therefore, owing to prescription of various antihyperglycemics and limitation of data retrieval, we are unable to analyze which medications may improve or worsen the survival rates.

Response to Reviewer #1’s comments:

1. We have deleted many sentences from the Introduction section in accordance with the reviewer’s advice since they are repeated in the Discussion. We completely agree with the reviewer’s suggestion. Further, we have edited some sentences for better clarity. 

We also agree with the reviewer’s concern regarding Line 82. We have included a supplementary table that summarizes the observed findings from references 15, 16, and 21-24 in the Supplementary section.

2. We are grateful for the reviewer’s concern for the potential threat of selection bias in the present study. In the Results section of our manuscript (under the subheading “Baseline characteristics of participants”), the following sentence was written: In total, 2242 lung resections for NSCLC were performed. A total of 23 patients had incomplete data; hence, data of 2219 (99.0%) patients were reviewed in the present study. 

3. As the reviewer pointed out, it is better to match exposed (DM) with unexposed (no DM) patients based on confounders such as age, gender, stage, and histological type. We completely agree with the reviewer. In the present study, first, we wanted to grasp the whole picture of individuals with DM and without. Therefore, we analyzed all the patients enrolled and then adjusted the confounders by statistical processing.

One more point the reviewer has raised is that the non-diabetic group included a 17-year-old patient. Checking the raw data, the patient underwent radical resection for NSCLC (adenocarcinoma). The data are correct. However, as pointed out in the section below, we have now presented continuous variables as medians and interquartile ranges (IQRs). Therefore, the number “17” has been deleted from the manuscript.

4. As the reviewer has suggested, we should clarify the timeframe for primary and secondary endpoints. Postoperative complications, as a primary endpoint, was assessed from the end of surgery to discharge from the thoracic surgery unit. Overall survival, as a secondary endpoint, was calculated from the initial event (date of surgery) to the final event (death or loss to follow-up). To clarify the timeframe for the endpoints, we have added a sentence in the Outcome Measures section. 

The original grading system for postoperative complications was published by Clavien et al. in 2004. The Japan Clinical Oncology Group (JCOG) has modified and updated the original Clavien-Dindo classification for more precise comparisons of the frequency and severity of postoperative complications among clinical trials across many different surgical fields. As we have cited in the manuscript (reference 25), this grading system for surgical complications has been frequently used and cited in clinical studies. Therefore, we consider this grading system to be valid enough to be used in this clinical study.

Grade was not included in the database. We created corresponding grades for every case, referring to the updated Clavien-Dindo classification.

5. (i) In accordance with the reviewer’s advice, we have changed the description to “presented as medians and interquartile ranges (IQRs).” We have accordingly, we have revised the tables.

 (ii) We have added “%” after the actual frequency for all categorical variables.

 (iii) The final event to stop the follow-up has been added to the sentence (Line 129 of manuscript with changes highlighted) for a better description of the operational definition of time to event.

 (iv) The reviewer pointed out that prediction was not the goal of this study and that the objective was to estimate the relationship between DM and overall survival. We completely agree with the reviewer’s comments, and unfortunately, our description regarding Cox regression was inappropriate. We performed Cox regression to ensure that DM is not an independent risk factor for survival after adjusting for potential confounders. Therefore, we have deleted the following sentence: “For cox regression, initial univariate comparisons were performed using relevant variables and those with an association yielding a p-value of <0.1 were put into the final models.” 

Further, we have used upper case “c” in cox.

 (v) The comparison of postoperative complications between individuals with and without DM is shown in Table 1. Hence, we would like to leave the number unchanged.

For the primary endpoint (postoperative complications), we have used the adjusted estimates as a binary event and added the adjusted estimates (odds ratios, 95% CI, and p-value) to Table 3.

By creating 4 groups (non-DM [reference], DM with HgA1c% <7%, DM with HgA1c% ≥7% to <8%, and DM with HgA1c% ≥8%), we have also used the adjusted estimates for the primary endpoint (postoperative complications). These data have been added to the manuscript as Table 3.

 (vi) The comparison of length of stay between individuals with and without DM is shown in Table 1. Hence, we would like to leave the number unchanged.

The Kaplan-Meier survival curves (Figs 1, 2, 3) have been adjusted for confounders (age and gender).

Table 3 has been renamed as Table 4 because an additional table was inserted after Table 2, which shows Cox regression analysis as a multivariable investigation. For more clarity, we have added “Multivariate” to the table title.

As pointed out by the reviewer, HbA1c% was measured in a very small group of non-DM subjects. We agree with the reviewer’s comment. We have deleted HbA1c% values from Table 4.

6. We have addressed all the points raised by the reviewer and have made the necessary revisions. Fortunately, the revisions have not drastically changed the whole picture. Therefore, we have left the Discussion section unchanged. 

Again, we would like to thank the editor and reviewer for the detailed review of our study. All comments have helped us improve the quality of our manuscript.

We await a favorable response and a new evaluation.

Sincerely,

Teruya Komatsu, MD

E-mail: tk.thoracic@gmail.com

---

## [Decision Letter · Decision Letter 1]

23 Oct 2020

Impact of diabetes mellitus on postoperative outcomes in individuals with non-small-cell lung cancer: a retrospective cohort study

PONE-D-20-19563R1

Dear Dr. Komatsu,

We’re pleased to inform you that your manuscript has been judged scientifically suitable for publication and will be formally accepted for publication once it meets all outstanding technical requirements.

Kind regards,

Yoshiaki Taniyama, MD, PhD

Academic Editor

PLOS ONE

Additional Editor Comments (optional):

Reviewers' comments:

Reviewer's Responses to Questions

**Comments to the Author**

1. If the authors have adequately addressed your comments raised in a previous round of review and you feel that this manuscript is now acceptable for publication, you may indicate that here to bypass the “Comments to the Author” section, enter your conflict of interest statement in the “Confidential to Editor” section, and submit your "Accept" recommendation.

Reviewer #1: All comments have been addressed

2. Is the manuscript technically sound, and do the data support the conclusions?

Reviewer #1: Yes

3. Has the statistical analysis been performed appropriately and rigorously? 

Reviewer #1: Yes

4. Have the authors made all data underlying the findings in their manuscript fully available?

Reviewer #1: Yes

5. Is the manuscript presented in an intelligible fashion and written in standard English?

Reviewer #1: Yes

6. Review Comments to the Author

Reviewer #1: (No Response)

7. PLOS authors have the option to publish the peer review history of their article (what does this mean?). If published, this will include your full peer review and any attached files.

Reviewer #1: **Yes: **Rita Popat

---

## [Editor Report · Acceptance letter]

29 Oct 2020

PONE-D-20-19563R1 

Impact of diabetes mellitus on postoperative outcomes in individuals with non-small-cell lung cancer: a retrospective cohort study 

Dear Dr. Komatsu:

I'm pleased to inform you that your manuscript has been deemed suitable for publication in PLOS ONE. Congratulations! Your manuscript is now with our production department. 

Kind regards, 

on behalf of

Dr Yoshiaki Taniyama 

Academic Editor

PLOS ONE